# A Low Concentration of Citreoviridin Prevents Both Intracellular Calcium Deposition in Vascular Smooth Muscle Cell and Osteoclast Activation In Vitro

**DOI:** 10.3390/molecules28041693

**Published:** 2023-02-10

**Authors:** Seongtae Jeong, Bok-Sim Lee, Seung Eun Jung, Yoojin Yoon, Byeong-Wook Song, Il-Kwon Kim, Jung-Won Choi, Sang Woo Kim, Seahyoung Lee, Soyeon Lim

**Affiliations:** 1The Interdisciplinary Graduate Program in Integrative Biotechnology & Translational Medicine (IBTM), Graduate School, Yonsei University, Seoul 03722, Republic of Korea; 2Institute for Bio-Medical Convergence, Catholic Kwandong University International St. Mary’s Hospital, Incheon 22711, Republic of Korea; 3Department for Medical Science, College of Medicine, Catholic Kwandong University, Gangneung-si 25601, Republic of Korea; 4Institute for Bio-Medical Convergence, College of Medicine, Catholic Kwandong University, Gangneung-si 25601, Republic of Korea

**Keywords:** vascular calcification, osteoporosis, citreoviridin

## Abstract

Vascular calcification (VC) and osteoporosis are age-related diseases and significant risk factors for the mortality of elderly. VC and osteoporosis may share common risk factors such as renin-angiotensin system (RAS)-related hypertension. In fact, inhibitors of RAS pathway, such as angiotensin type 1 receptor blockers (ARBs), improved both vascular calcification and hip fracture in elderly. However, a sex-dependent discrepancy in the responsiveness to ARB treatment in hip fracture was observed, possibly due to the estrogen deficiency in older women, suggesting that blocking the angiotensin signaling pathway may not be effective to suppress bone resorption, especially if an individual has underlying osteoclast activating conditions such as estrogen deficiency. Therefore, it has its own significance to find alternative modality for inhibiting both vascular calcification and osteoporosis by directly targeting osteoclast activation to circumvent the shortcoming of ARBs in preventing bone resorption in estrogen deficient individuals. In the present study, a natural compound library was screened to find chemical agents that are effective in preventing both calcium deposition in vascular smooth muscle cells (vSMCs) and activation of osteoclast using experimental methods such as Alizarin red staining and Tartrate-resistant acid phosphatase staining. According to our data, citreoviridin (CIT) has both an anti-VC effect and anti-osteoclastic effect in vSMCs and in Raw 264.7 cells, respectively, suggesting its potential as an effective therapeutic agent for both VC and osteoporosis.

## 1. Introduction

Defined as extracellular deposition of calcium-phosphate complexes in the arterial wall, vascular calcification (VC) is known to be an independent predictor of cardiovascular disease-related mortality [1,2,3]. VC once was considered as a passive process, but now it is regarded as an active and regulated pathological process, where vascular smooth muscle cells (vSMCs) play an important role [4]. In stimulated vascular walls, vSMCs can trans-differentiate into an osteoblast-like phenotype by the changes of various VC stimulating factors and/or inhibitors involved in bone metabolism [5].

Osteoporosis is characterized by demineralization of bone tissue caused by the imbalance between the osteoblast-mediated bone formation and the osteoclast-mediated resorption, and the resultant increase of bone fragility increases a risk of fracture in return [6]. Pro-inflammatory cytokines, such as tumor necrosis factor alpha (TNF-α), interleukin-1 (IL-1), IL-6, and receptor activator of nuclear factor-kappa B ligand (RANKL), are known to increase bone resorption by stimulating osteoclast differentiation and activation, and consequently, inhibit bone formation [7]. Interestingly, these bone metabolic mediators have been associated with the development of VC as well [8,9].

Since the late 1990s, clinical and laboratory findings have demonstrated the association between osteoporosis and vascular calcification, and it also has been reported that they share a number of common risk factors such as aging, estrogen deficiency, chronic inflammation, oxidative stress, and lipid metabolism [10,11,12,13,14]. Furthermore, it has been suggested that, especially under hypertensive conditions, vascular calcification and osteoporosis share pathophysiological mechanisms involving the renin-angiotensin system (RAS) [15].

Angiotensin II (Ang II)-induced excessive reactive oxygen species (ROS) can activate Src-family kinases, protein kinase C (PKC), and mitogen-activated protein kinases (MAPKs) in vSMCs [16]. This signaling cascade, in turn, activates a series of transcription factors responsible for osteogenic gene expressions, such as nuclear factor kappa-B (NF-κB), Runx2, and activator protein 1 (AP-1) contributing to VC [17,18,19]. In addition to directly inducing osteoblast-like differentiation of vSMCs leading to VC, Ang II can also indirectly contribute to bone resorption by activating osteoclast via secretion of RANKL.

Upon binding to RANK, RANKL recruits adaptor molecules such as tumor necrosis factor receptor-associated factor 6 (TRAF6), and this sequentially activates MAPKs, NF-κB, and AP-1. In turn, the activated NF-κB induces the expression of nuclear factor of activated T-cells cytoplasmic 1 (NFATc1), a major osteoclastogenesis regulator [20,21]. Furthermore, NFATc1 also increases a number of genes including tartrate-resistant acid phosphatase (TRAP), Cathepsin K (CTSK), calcitonin receptor (CALCR), and dendritic-cell-specific transmembrane protein (DC-STAMP) to regulate osteoclast differentiation [22,23], leading to the formation and activation of osteoclasts.

It also has been demonstrated that the expression of RANKL can be increased in calcifying vSMCs in response to oxidative stress [24], and Ang II induced expression of RANKL in human vSMCs and in ApoE knockout mice has been demonstrated as well [25]. Additionally, Ang II can also induce the expression of RANKL and subsequent extracellular secretion in non-vSMCs such as osteoblasts or synovial cells indirectly contributing to osteoclast activation and osteoporosis. [26,27,28].

As circumstantial evidence of the involvement of RAS in both VC and osteoporosis, preclinical studies have demonstrated that Angiotensin type 1 receptor blockers (ARBs) have a protective effect on vascular calcification [25,29]. In addition, a beneficial effect of ARBs on hip fracture in elderly has been demonstrated in a cohort study [30]. However, upon a closer look, this particular cohort study also demonstrated that there may a sex-dependent discrepancy in the responsiveness to ARB treatment.

To be more specific, the risk of fracture was more decreased in men than women following ARB treatment, and especially older women in postmenopausal period showed much higher risk of fracture [30]. Considering estrogen deficiency can lead to a prolonged bone loss by promoting osteoclast formation and life span [31,32], the reported sex-dependent discrepancy may stem from the estrogen deficiency in older women. This suggested that blocking Ang II signaling pathway may not be effective to suppress bone resorption especially if an individual has underlying osteoclast activating conditions such as estrogen deficiency. Therefore, it has its own significance to find alternative chemical agents to inhibit the development of both VC and osteoporosis, possibly by directly targeting osteoclast activation to circumvent the shortcoming of ARBs in preventing bone resorption in estrogen deficient individuals.

In the present study, a natural compound library was screened to find chemical agents that are effective in preventing both calcium deposition in vSMCs and activation of osteoclast using experimental methods such as Alizarin red staining (calcium deposition) and TRAP staining (osteoclast activation). According to our data, citreviridin (CIT) has both an anti-VC effect and anti-osteoclastic effect in vSMCs and in Raw 264.7 cells, respectively, suggesting its potential as an effective therapeutic agent for both VC and osteoporosis.

## 2. Results

### 2.1. CIT Attenuates Both Ang II-Induced VC and RANKL-Induced Osteogenic Differentiation

Using a natural product library composed of 502 natural compounds, primary screening for agents (2 μg/mL, each) that could suppress calcium deposition in vSMCs was conducted. A selection threshold was ≥70% of calcium deposition inhibition, and 21 candidate compounds were selected (Figure 1). As a secondary screening, the selected compounds were further screened for their anti-osteoclastic effect using RANKL-treated Raw264.7 cells. The results indicated that six compounds (Figure 2) showed the protective effect. However, CIT was the only compound that has never been reported to be effective on VC and/or osteoporosis and, thus, the effect of CIT on both VC and osteoporosis was further examined for this study.

### 2.2. Effect of CIT on vSMC Viability

Upon morphological examination of vSMCs treated with CIT (1 and 2 μg/mL) with or without Ang II (500 nM), 2 μg/mL of CIT showed a mild cytotoxicity (Figure 3A and 3B), which was not observed during the first screening. Since the major difference in the protocols used for the first screening and this cytotoxicity testing was the size culture dishes (48 well vs. 60 mm, respectively), this unexpected cytotoxicity was possibly due to the absolute amount of CIT used for this experiment (0.8 μg for 48 well plates vs. 4 μg for 60 mm plates). Nevertheless, 1 μg/mL of CIT did not show any significant decrease of viability (Figure 3C) or cytotoxicity (Figure 3D).

### 2.3. CIT Inhibits Ang II-Induced Calcium Deposition in vSMCs

First, Alizarin red S staining indicated that Ang II treatment apparently increased calcium deposition in vSMCs, and this was suppressed by CIT treatment (Figure 4A). Quantification of Alizarin red S showed that 1 μg/mL of CIT significantly decreased the amount of calcium deposited in vSMCs compared to the Ang II treated control (4.91 ± 1.58 fold increase vs. 1.46 ± 0.19 fold increase compared to the untreated control). Lower concentrations of CIT (0.1 and 0.5 μg/mL) also inhibited Ang II-induced calcium deposition, and it showed a concentration-dependent tendency of decrease (Figure 4B). However, additional calcium assay results indicated that only 1 μg/mL of CIT was significantly effective (Figure 4C) and, therefore, 1 μg/mL of CIT was used for the rest of the experiments.

### 2.4. CIT Attenuates Ang II-Induced Osteogenic Marker Expressions in vSMCs

According to our Western blot data, Ang II significantly decreased the expressions of SMC differentiation markers, namely Calponin and smooth muscle protein 22-alpha (SM22α), but this was attenuated by CIT (Figure 5A). On the other hand, Ang II significantly increased the expression of Osterix, an osteoblast marker, at both protein and mRNA level, and this was also attenuated by CIT (Figure 5B). Additionally, Ang II significantly increase the mRNA expression of ALP, an osteoblast activity marker, and this was again significantly suppressed by CIT (Figure 5C). The amount of ALP secreted from the cells was estimated by the ALP activity of culture media, and the results showed that CIT was able to significantly suppress the Ang II-induced ALP secretion from vSMCs (Figure 5D).

### 2.5. CIT Inhibits Ang II-Induced ROS Production in vSMCs

Since excessive ROS can lead to VC and, thus, a ROS-lowering agent can prevent VC [33,34], the effect of CIT on Ang II-induced ROS production in vSMCs was examined. Flow cytometry data indicated that Ang II significantly increased ROS production in vSMCs, but it was significantly attenuated by CIT (Figure 6A). As modulators of ROS production, the expressions of NADPH oxidase (NOX) and phosphorylation status of PKC and Src were examined. As shown in the Figure 6B, the expression of NOX was significantly increased by Ang II, but this was significantly attenuated by CIT. Ang II also significantly increased the phosphorylation of PKCδ and Src (Tyr416), which was abrogated by CIT (Figure 6C).

### 2.6. CIT Inhibits Ang II-Induced MAPK Signaling and Subsequent NF-κB and AP-1 Activation

According to our data, Ang II significantly increased the phosphorylation of MAPKs (extracellular signal-regulated kinase (ERK), c-Jun N-terminal kinase (JNK) and p38). Although CIT showed no significant inhibitory effect on the Ang II-induced ERK phosphorylation, it did significantly inhibit the Ang II-induced phosphorylation of JNK and P38 (Figure 7A). Furthermore, CIT significantly suppressed the Ang II-induced phosphorylation of downstream transcription factors, namely NF-κB (Figure 7B) and AP-1 (Figure 7C).

### 2.7. CIT Inhibits RANKL-Induced Osteoclast Differentiation of Raw264.7 Cells

First, any possible cytotoxicity of CIT on Raw264.7 cells was evaluated by using LDH assay and Annexin V/PI staining. CIT up to 1 μg/mL did not show any significant cytotoxic effect on Raw264.7 cells (Figure 8A), and Annexin V/PI staining also indicated that 1 μg/mL of CIT did not cause apoptosis in Raw264.7 cells (Figure 8B). The results of TRAP staining to visualize the osteoclast marker enzyme demonstrated that RANKL apparently increased the number of TRAP positive cells, while CIT abrogated the pro-osteoclastic effect of RANKL (Figure 9A). Quantification of TRAP staining indicated that CIT significantly suppressed RANKL-induced osteoclast differentiation of Raw264.7 cells as evidenced by the decrease of both TRAP-positive cells (Figure 9B) and TRAP-positive multinuclear cells (Figure 9C,D) in a concentration-dependent manner. Furthermore, RANK increased the mRNA expressions of osteogenic markers such as TRAP, CTSK, CALCR, DC-STAMP, and NFATc1, but these were significantly abrogated by CIT (Figure 9E).

### 2.8. CIT Inhibits RANKL-Induced Nuclear Translocation of NFκB and NFATc1

According to our data, RANKL significantly increased both the phosphorylation of NF-κB (Figure 10A) and expression of NFATc1 (Figure 10C). However, CIT significantly abrogated the RANKL-induced increase of NF-κB phosphorylation and NFATc1 expression. Furthermore, immunocytochemistry using NF-κB (Figure 10B) and NFATc1 (Figure 10D) specific antibodies demonstrated that CIT also significantly suppressed the RANKL-induced nuclear translocation of these transcription factors in Raw264.7 cells.

### 2.9. CIT Inhibits RANKL-Induced Osteoclast Activity

The results of Pit formation assay indicated that RANKL significantly increased the bone resorptive activity of Raw264.7 cells (Figure 11A), but this was significantly suppressed by CIT reducing both Pit diameter (Figure 11B) and area (Figure 11C).

### 2.10. Ang II-Induced RANKL Secretion from vSMCs May Connect VC and Osteoporosis

To examine the possibility that RANKL being a mediator linking VC and osteoporosis, the expression of RANKL in vSMCs following Ang II stimulation was examined. As shown in the Figure 12A, Ang II increased the mRNA expression of RANKL in vSMCs, and this was attenuated by CIT. Furthermore, the amount of RANKL protein presented in the culture medium of Ang II-stimulated vSMCs was increased compared to that in the untreated control (Figure 12B), suggesting that the RANKL released from the vSMCs upon Ang II stimulation may, in turn, promote the activation of osteoclasts leading to the development of osteoporosis.

## 3. Discussion

In the present study, we report that a low concentration of natural compound CIT effectively suppressed the Ang II-induced calcium deposition in HAoSMCs and RANKL-induced osteoclastogenesis of Raw264.7 cells. However, its reported toxicity can be an issue if it were to be further evaluated for its potential as a therapeutic agent for both VC and osteoporosis.

CIT, a toxic secondary metabolite derived from Penicillium strains, has been associated with the development of cardiovascular diseases such as atherosclerosis and cardiac beriberi [35,36], and its biological effects including cytotoxicity have been evaluated in different in vitro and in vivo systems. For example, disruption of nerve and muscle metabolism by CIT was reported to be responsible for the development of beriberi [36], and it is also known to cause Keshan disease through oxidative stress [37]. Furthermore, CIT induced autophagic cell death through the lysosomal–mitochondrial axis in hepatocytes [38]. Especially pertaining to cardiovascular system, it has been reported that CIT enhanced tumor necrosis factor-α (TNFα)-induced endothelial adhesion by increasing the expression of adhesion molecules such as ICAM-1, VCAM-1, and E-selectin in human umbilical vein endothelial cells [39], and it caused myocardial apoptosis via activating autophagic pathway [40]. However, currently available empirical data on the cytotoxicity of CIT is just not sufficient to draw an exclusive conclusion on CIT’s cytotoxicity, and it needs to be carefully examined, especially in terms of the concentration used for the experiments.

First of all, there are not that many in vitro studies examined the cytotoxicity of CIT over a range of varying concentrations to which our data can be compared, not to mention in vitro studies that used vSMCs or Raw264.7 cells. For example, the above-mentioned study examined the CIT-induced autophagy-dependent apoptosis of hepatocytes used 5 μM of CIT throughout the study [38], and yet another study reported CIT-induced myocardial apoptosis through autophagic pathway used 0.1–0.3 mg/kg of CIT to demonstrate myocardial apoptosis in vivo, but it did not directly show the cytotoxicity of CIT on in vitro cultured cardiomyocytes. Although they used 2–8 μM of CIT for some in vitro experiments, those experiments were mainly to examine the effect of CIT on autophagy in cardiomyocytes rather than on cell viability [40]. On the other hand, although the cell used was porcine kidney cells, a concentration–response curve for CIT was generated in one study, and the results indicated that even up to 10^1.5^ μM of CIT (approximately 32 μM) did not show any significant cytotoxicity in MTT (3-(4,5-dimethylthiazol-2-yl)-2,5-diphenyltetrazolium bromide, a tetrazole) assay [41].

In the present study, 1 μg/mL of CIT was used for most of the experiments, and it is approximately equivalent to 2.5 μM of CIT (molecular weight of CIT is 402.5). More importantly, this particular concentration of CIT did not show any significant cytotoxic effect on both vSMCs and Raw264.7 cells, although 2 μg/mL of CIT did show mild cytotoxicity (Figure 3). Certainly, this does not necessarily mean that a low concentration of CIT cleared the issue of possible cytotoxicity. However, at this point, our data simply indicated that, similar to other potentially toxic substances used for therapeutic purposes such as Botulinum toxin [42], CIT may do more good than harm at certain given concentrations. Therefore, it will be worthwhile to systemically examine the effects of varying concentrations of CIT, especially on VC and osteoporosis, to verify the narrow therapeutic window of CIT on VC and osteoporosis that may or may not exist in further in vivo studies.

Prior to the development of VC, a phenotypic change of vSMCs from contractile to osteogenic occurs accompanying the down-regulation of vSMC contractile markers such as smooth muscle α-actin and up-regulation of osteogenic markers such as ALP [43]. Our data demonstrated that CIT significantly suppressed Ang II-induced calcium deposition in vSMCs (Figure 4) and osteogenic trans-differentiation of vSMCs (Figure 5), indicating CIT can attenuate calcium deposition in vSMCs by preventing phenotype switching of vSMCs.

In a mechanistic point of view, elevated ROS production has been associated with VC leading to activation of osteogenic signaling pathways [33]. Ang II is well known to induce excessive production of ROS [44], and the Ang II-induced VC in the present study might as well be the result of the excessive ROS produced by Ang II. Therefore, it is reasonable to assume that inhibition of ROS production can be an effective approach to attenuate phenotype switching of vSMCs and resultant VC. Agreeing with such assumption, CIT significantly suppressed Ang II-induced production of ROS in the present study (Figure 6). In addition, at transcriptional level, osteogenic transcriptional factors activated via MAPK signaling pathway are known to drive the transcription of osteogenic genes such as Runx2 and Osterix [45]. Our data also indicate that CIT significantly suppressed the Ang II-induced phosphorylation of JNK and p38, and subsequent activation of the osteogenic transcription factor NFκB and AP-1 (Figure 7). These data suggested that CIT can be an agent that effectively suppress the Ang II-induced VC in vSMCs. One of the main cause of osteoporosis, the other pathologic condition on which the effect of CIT was examined in the present study, is pathologically enhanced activity of osteoclast and resultant increase of bone resorption [46]. It is a well-established fact that monocyte/macrophage lineage precursor cells can differentiate into osteoclasts by RANKL [47], and RANKL-induced osteoclast differentiation of Raw264.7 cell was also demonstrated in the present study. However, such RANKL-induced osteoclast differentiation was significantly abrogated by CIT, along with down-regulated expression of osteoclast-specific genes such as TRAP, CTSK, CALCR, and DC-STAMP (Figure 9) and attenuated translocation of NF-κB/NFATc1 signaling (Figure 10). DC-STAMP is known to play an important role in cell–cell fusion to create multi-nucleated cells [48], and TRAP, CALCR, and CTSK are the acidic substances secreted by mature osteoclasts contributing to the bone resorptive activity by degrading bone surface [49,50]. Therefore, it was speculated that the activity of osteoclast could be suppressed by CIT, and our data on bone resorption assay strongly supported such speculation (Figure 11).

Although our results demonstrated that CIT could effectively inhibit both Ang II-induced osteogenic differentiation of vSMCs and RANKL-induced osteoclast differentiation of Raw264.7 cells so far, those data were obtained from two independent in vitro systems and, thus, not sufficient enough to claim that CIT will simultaneously act on both VC and osteoporosis in a single entity. Therefore, as a possible missing link between these two systems, the expression of RANKL production from Ang II-stimulated vSMCs was examined in the present study. In fact, Ang II has been reported to increase the expression of RANKL in vSMCs [25], and it can promote macrophage migration and differentiation into osteoclast-like cells [24]. Our data also indicated that Ang II significantly increased mRNA expression of RANKL, which was attenuated by CIT, and secretion of RANKL as well (Figure 12). Such a role of RANKL as a linker that connects VC and osteoporosis has been demonstrated in a previous study where PKA agonist-treated aortic SMCs were co-cultured with Raw264.7 cells that eventually differentiated into osteoclast by the RANKL secreted by the aortic SMCs [51].

Additionally, Ang II stimulation is known to induce a variety of pro-inflammatory factors, such as interleukin 1β (IL-1β), IL-6, tumor necrosis factor-α (TNF-α), and high mobility group box 1 protein (HMGB1) from various cells [52,53,54]. In turn, these factors can affect the expression of RANKL. For example, extracellular HMGB1 can enhance the expression of RANKL that can act as a ligand for toll-like receptors (TLRs) and the receptor for advanced glycation end products (RAGE) in osteoblastogenic bone marrow stromal cell cultures, osteocytes, and osteoblasts [55,56]. Therefore, a crosstalk between VC and osteoporosis can be facilitated by many different pathways and cell types involving production of RANKL in vivo. Nevertheless, since CIT worked on RANKL-induced osteoclast differentiation of Raw264.7 cells even without the presence of Ang II, CIT is expected to work on both pathologic conditions in a single entity, as long as they are mainly linked by RANKL.

## 4. Materials and Methods

### 4.1. Cell Culture

#### 4.1.1. Culture and Calcium Deposition of Human Aortic Smooth Muscle Cells (HAoSMCs)

HAoSMCs were purchased from Lifeline cell technology (Frederick, MD, USA; FC-0015) and cultured in VascuLife^®^ Basal Medium (LM-0002; Lifeline) with supplements (VascuLife^®^ SMC LifeFactors kit; LS-1040; Lifeline). For antibiotics, 100 U/mL penicillin (15140-122; Thermo Fisher Scientific, Waltham, MA, USA) and 100 μg/mL streptomycin (15140-122; Thermo Fisher Scientific) were used.

To induce vascular calcification or calcium deposition, cells of passage 6–8 were grown in a 48-well culture plate at density of 6 × 10^4^ cells per wells in a HAoSMC culture medium. After cells were attached, the medium was changed to Ang II-based calcification induction media; mixture of Ang II (500 nM), β-Glycerophosphate (β-GP; 4 mM), CaCl_2_ (3.8 mM) and 15% fetal bovine serum (FBS) in the Basal medium. The cells were cultured for 6–8 days for calcium deposition.

#### 4.1.2. Culture and Osteoclast Differentiation of Raw264.7 Cells

Raw264.7 cells (TIB-71; ATCC, Manassas, VA, USA), an established macrophage cell line for osteoclastic differentiation, were cultured in Dulbecco’s modified Eagle’s medium (DMEM) (30-2002; ATCC) with 10% FBS (16000-044, Thermo Fisher Scientific, Waltham, MA, USA), 100 U/mL penicillin (15140-122, Thermo Fisher Scientific), and 100 μg/mL streptomycin (15140-122; Thermo Fisher Scientific) at 37 °C in a humidified incubator under 5% CO_2_ atmosphere. For cell differentiation and drug treatment, cells were used between passage 5 and 15.

To differentiate osteoclast cells from Raw264.7 cells, the cells were seeded on 48-well plates (3 × 10^3^ cells/well) in complete DMEM media. After cell attachment, complete DMEM medium was changed to complete alpha modified minimal essential medium (α-MEM; 11900-024, Thermo Fisher Scientific) as differentiation medium. Raw264.7 cells were treated 40 ng/mL recombinant RANKL (ALX-522-131; ENZO Life Science, Farmingdale, NY, USA) every 2 days for 4 days.

### 4.2. Screening of Natural Compounds for Suppressing Calcium Deposition of Human Aortic Smooth Muscle Cells (HAoSMCs) Using Alizarin Red Staining

To screen natural compounds for suppressing Ang II-induced calcium deposition in HAoSMCs, ENZo library was used (Screen-well^®^ Natural product library, BML-2865). Natural compounds were added upon induction with Ang II. Intracellular calcium deposition was assessed by alizarin red staining. Briefly, the cells were fixed in cold 70% ethanol for 30 min at 4 °C. The fixed cells were stained with 2% alizarin red S (A5533; Sigma-Aldrich, Seoul, Republic of Korea) for 10 min at room temperature. The cells were observed under the microscope (CKX41; Olympus, Tokyo, Japan) and images were obtained by a digital camera (eXcope T300; Olympus) at 40× magnification. For quantification, the stained cells were destained in 10% cetylpyridinum chloride (C0732; Sigma-Aldrich). The absorbance was measured at 595 nm using a microplate reader (Multiskan FC; Thermo Fisher Scientific).

### 4.3. Evaluation of Cell Viability (WST-1) and Toxicity (LDH)

Cell viability was assessed by water-soluble tetrazolium 1 (WST-1) assay. The Cell viability was checked using EZ-Cytox water-soluble tetrazolium salt (WST) assay kit (EZ-3000; Dogenbio, Seoul, Republic of Korea). The absorbance was measured at 450 nm by microplate reader (Multiskan FC; Thermo Fisher Scientific). Cytotoxicity was evaluated by using Lactate dehydrogenase (LDH) Cytotoxicity Detection Kit (MK401; TAKARA, Otsu, Japan) according to the manufacturer’s protocols. Briefly, a total of 100 μL of LDH reaction reagent was added into 100 μL of supernatants and then incubated at 37 °C for 30 min. Absorbance was determined at 450 nm using microplate reader (Multiskan FC; Thermo Fisher Scientific).

### 4.4. Annexin V/PI Staining

To assess cell death, Annexin V/propodium iodide (PI) staining was performed using FITC Annexin V Apoptosis Detection Kit I (556547; BD biosciences, Becton, NJ, USA). Briefly, the cells were stained with 5 μL of Annexin V and 5 μL of PI at room temperature for 15 min. The cells were diluted with the binding buffer and analyzed with BD Accuri™ C6 flow cytometer (BD biosciences).

### 4.5. Calcium Assay

To quantify the amount of calcium deposited, HAoSMCs were incubated overnight with 0.6 N HCl at 4 °C to dissolve the deposited calcium. The amount of calcium present in the supernatants was determined by using QuantiChrom™ Calcium Assay Kit (DICA-500; Bioassay systems, Hayward, CA, USA). Briefly, a total of 200 μL of working reagent mixture was added into 3 μL of supernatants and then incubated at room temperature for 3 min. Absorbance was determined at 612 nm using microplate reader (Multiskan FC; Thermo Fisher Scientific).

### 4.6. Alkaline Phosphatase (ALP) Assay

Osteoblast-like differentiation of HAoSMCs was evaluated by using an alkaline phosphatase assay kit (ab83369; Abcam, Cambridge, UK) following the manufacturer’s protocol. The absorbance of supernatants was determined at 405 nm using a microplate reader (Multiskan FC; Thermo Fisher Scientific).

### 4.7. Intracellular ROS Detection

For reactive oxygen species (ROS) detection, the cells were harvested with accutase (A6964; Sigma-Aldrich). A total of 5 × 10^5^ cells in cell suspension were incubated with 5 µM CM-H2DCFDA in the dark for 10 min at 37 °C. The cell suspension was collected by centrifugation at 1600× *g* rpm and supernatants were removed. Collected HAoSMCs were resuspended with 500 µL pre-warmed medium. Intracellular ROS levels in HAoSMCs were then immediately determined via BD Accuri™ C6 flow cytometer (BD biosciences). The result was calculated from four independent experiments and analyzed as fold change compared to control.

### 4.8. TRAP Staining

Tartrate-Resistant Acid Phosphatase (TRAP) staining was performed using the TRACP & ALP double-stain Kit (MK300; TAKARA) according to the manufacturer’s instructions. The cells were gently washed with PBS. A total of 120 μL of fixation solution was added to each well for 5 min at room temperature. After that, the cells were washed using sterile distilled water (D.W.). A total of 120 μL of substrate solution was added to each well and covered with parafilm to prevent drying. The plate was incubated at 37 °C for 45 min and washed three times with D.W. to stop the reaction. TRAP-positive cells were visualized and captured via microscope (CKX41; Olympus, Tokyo, Japan) and digital camera (eXcope T300; Olympus). Osteoclast cells that stained as a purple color were counted as TRAP-positive cells from three or more independent experiments. The number of the TRAP-positive cells that have three more nucleus was measured by NIH ImageJ 1.52a software (Silk Scientific Corp., Orem, Utah).

### 4.9. Pit Assay

The bone resorption assay kit (CSR-BRA-48KIT; COSMOBIO, Tokyo, Japan) was used to check resorption activity of osteoclast cells according to the manufacturer’s protocol. Briefly, Raw264.7 cells (5 × 10^3^ cells/well) were seeded into calcium phosphate (CaP)-coated 48-well culture plate. Additional collagen type I coating was performed using 50 μg/mL collagen type I (3447-020-01; R&D System, Minneapolis, MN, USA) to mimic a bone biomimetic surface. Raw264.7 cells were cultured at 37 °C and 5% CO_2_ in DMEM containing 10% FBS. After 24 h, the medium was changed to complete α-MEM medium without phenol-red and then pretreated with a vehicle (DMSO) and citreoviridin (0.5 and 1μg/mL) for 1 h. Subsequently, RANKL (100 ng/mL) was added in the medium to induce an osteoclast differentiation. RANKL and citreoviridin were re-treated every 2 days for 6 days. On day 6, the cells were washed with PBS and treated with 5% sodium hypochlorite for 5 min to remove the cells. Then, the plate was washed with D.W. and dried. Osteoclast resorbing areas were captured using a digital camera (Olympus) attached to microscope (Olympus) and the pit areas obtained from 20 different regions by 4 independent experiments were measured by NIH ImageJ 1.52a software.

### 4.10. Reverse Transcription PCR (RT-PCR)

For RT-PCR, total RNA was isolated from the cells using Hybrid-R (305-101; GeneAll Biotechnology, Korea) according to manufacturer’s protocol. The amount of RNA was measured by Nanodrop one (Thermo Fisher Scientific), and 1 μg of RNA was used to synthesize cDNA using the Maxime RT PreMix Kit (25081; iNtRON Biotechnology, Seongnam, Korea). For PCR reaction, AccuPower^®^ PCR PreMix (K-2016; Bioneer, Daejeon, Korea) was used. PCR was performed under the following conditions using a PCR machine (C1000 touch Thermal cycler; Bio-Rad, Hercules, CA, USA): denaturation at 95 °C for 20 s, annealing at 56 °C for 30 s, and extension at 72 °C for 30 s for 35 cycles and, final extension at 72 °C for 5 min. The human (h) and mouse (m) specific primers used are the followings:

(m)Calcitonin receptor (forward: 5′-AGC TTG TTG GCA CTT TGT AT-3′; reverse: 5′-TTG CCT ATG CCA GGA CCA AT-3′), (m)Cathepsin K (forward: 5′-GCA GAT GTT TGT GTT GGT CTC T-3′; reverse: 5′-TGG TGG AAA GGT GTG ACA GG-3′), (m)DC-STAMP (forward: 5′-TTG AAC CGA GCT GCA TTC CT-3′; reverse: 5′-GCA CTA CCT TGG CCT TAC CT-3′), (m)NFTAc1 (forward: 5′-GGA GAG TCC GAG AAT CGA GAT-3′; reverse: 5′-TTG CAG CTA GGA AGT ACG TCT -3′), (m)TRAP (forward: 5′-CTC CTG CCT GTT CTC TTC CCA-3′; reverse: 5′- AAG AGA GAA AGT CAA GGG AGT GGC-3′), (m)GAPDH (forward: 5′-CAA GGT CAT CCA TGG ACA ACT TTG-3′; reverse: 5′-GTC CAC CAC CCT GTT GCT GTA G-3′).

(h)RANKL (118bp; forward: 5′-CCC AAG TTC TCA TAC CCT GAT G-3′; reverse: 5′-TTC CTC TCC AGA CCG TAA CT-3′), (h)ALP (108bp; forward: 5′-ATG GGA TGG GTC TCC ACA-3′; reverse: 5′-CCA CGA AGG GGA ACT TGT C-3′), (h)Osterix (711bp; forward: 5′- GCT TGA GGA GGA AGT TCA CTA T-3′; reverse: 5′-CCT TCT AGC TGC CCA CTA TTT-3′), (h)GAPDH (133bp; forward: 5′-CAT GGG TGT GAA CCA TGA GA-3′; reverse: 5′-GGT CAT GAG TCC TTC CAC GA-3′).

The PCR products from 3 or more independent experiments were electrophoresed on a 1.5% agarose gel. The gel images were captured by a BioRad ChemiDoc XRS imaging system (Bio-Rad) and analyzed with NIH ImageJ 1.52a software. The relative expression in each gene was then calculated using Glyceraldehyde 3-phosphate dehydrogenase (GAPDH).

### 4.11. Western Blot

Proteins were extracted from the cells using RIPA buffer (25 mM Tris pH 7.6, 150 mM NaCl, 1% NP-40, 1% sodium deoxycholate, 0.1% SDS) containing protease inhibitor (sc-11697498001; Santa Cruz Biotechnology, Inc., Dallas, TX, USA) and phosphatase inhibitors (4906845001; Thermo Fisher Scientific). The protein concentration was determined using the Bicinchoninic Acid Assay (BCA) method. The protein samples were separated on 12% sodium dodecyl sulfate (SDS)-polyacrylamide gel and then transferred to immobilon-P PVDF membranes (IPVH00010; Merk Millipore, Burlington, MA, USA). The membranes were blocked with 5% skim milk for 30 min at room temperature and incubated overnight at 4 °C with the following antibodies; ERK (1:1000; Cell signaling Technology, Danver, MA, USA, 9102), phospho-ERK (1:1000; Santa Cruz, sc-7383), PKCδ (1:1000; Cell signaling, 2085), phospho-PKCδ (1:500; Cell signaling, 9374), Src (1:1000; Cell signaling, 2110), phospho-Src (1:1000, Cell signaling, 2101), p38 (1:1000; Cell signaling, 9212), phospho-p38 (1:1000; Cell signaling, 9211), JNK (1:1000; Cell signaling, 9252), phospho-JNK (1:1000; Cell signaling, 9251), NF-κB p65 (1:1000; Cell signaling, 6956), phospho-NF-κB p65 (1:1000; Cell signaling, 3033), c-jun (1:1000, Cell signaling, 9165), phospho-c-jun (1:500; Cell signaling, 2361), SM22α (1:5000; Abcam, ab14106), Calponin (1:1000; Abcam, ab46794), α-SMA (1:1000, Abcam, ab5694), osterix (1:500, Santa Cruz, sc393325), RANKL (1:1000; Santa Cruz, sc377079), GAPDH (1:5000; Santa Cruz, sc32233), beta actin (1:5000; Santa Cruz, sc47778).

For secondary antibodies, the following secondary antibodies were used; anti-mouse (ADI-SAB-100J; ENZO) or anti-rabbit (ADI-SAB-300-J; ENZO). Secondary antibodies were used at 1:2000 dilution in 5% skim milk in TBST for 1 h at room temperature. Developed using the enhanced chemiluminescence method (AbClon, Seoul, Republic of Korea) and then detected protein expression using a BioRad ChemiDoc XRS imaging system (Bio-Rad). The protein bands were quantified by ImageJ 1.52a software. To detect a soluble RANKL level from HAoSMCs, culture media was concentrated by 100x using Vivaspin^®^ Turbo 4 (VS04T91; Satorius, Goettingen, Germany).

### 4.12. Immunocytochemistry

To determine of NF-κB p65 and NFATc1 translocation, immunocytochemistry was performed. After treatment, the cells were rinsed with PBS and fixed with 4% paraformaldehyde in PBS for 10 min at room temperature. The cells were then permeabilized with 0.2% Triton X-100 in PBS for 10 min at room temperature. Nonspecific staining was blocked by 2.5% normal horse serum blocking solution (S-2012-50; Vector Laboratories, Burlingame, CA, USA) for 10 min. After blocking, the cells were incubated with primary antibodies against total NF-κB (3039; Cell Signaling) or NFATc1 (sc-7294; Santa Cruz). All antibodies were diluted 1:200 in 2.5% normal horse serum for overnight at 4 °C. Rhodamine-conjugated anti mouse (1:500; AP124R, Merk Millipore) secondary antibodies were used. The cell nuclei were stained with diamidino-2-pehnylindole (DAPI) (D21490; Thermo Fisher Scientific). The images of NF-κB and NFATc1 translocation were taken using a LSM 700 laser scanning confocal microscope (Carl zeiss, Overkochen, Germany).

### 4.13. Statistical Analysis

The data were expressed as the mean ± standard error of the mean (SEM). Statistical analyses were performed using GraphPad Prism 7 software (GraphPad Software, San Diego, CA, USA). One-way analysis of variance (ANOVA) was used to compare three or more groups followed by a Bonferroni post hoc test. Student’s t test was used to compare two groups. Differences with *p* values of less than 0.05 were considered statistically significant.

## 5. Conclusions

In the present study, a novel biological effect of CIT on VC and osteoporosis that has not been reported anywhere has been demonstrated. Our data suggest that CIT can be an effective agent to simultaneously control both VC and osteoporosis and call for further studies to validate its in vivo efficacy and to find an optimal therapeutic window.

## Figures and Tables

**Figure 1 molecules-28-01693-f001:**
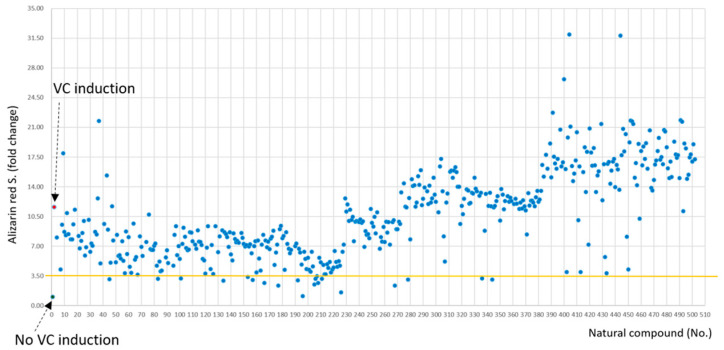
Functional screening of 502 natural compounds inhibiting in vitro calcium deposition. Compounds were co-treated with Angiotensin II (Ang II)-based calcification induction media in human aortic smooth muscle cells (HAoSMCs) for 8 days and then analyzed with Alizarin red staining. Red dot means an Ang II-based calcification induction group.

**Figure 2 molecules-28-01693-f002:**
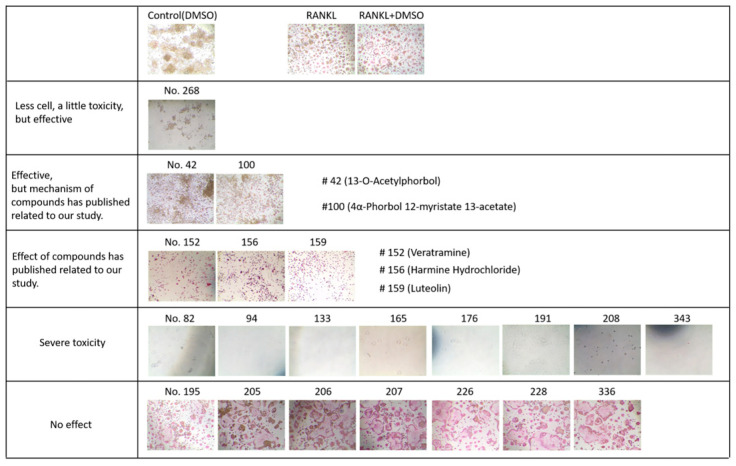
A total of 21 candidate compounds was further screened on their anti-osteoclastic effect in vitro. The compounds was co-treated with RANKL in Raw264.7 cells for 4 days and then analyzed with TRAP staining.

**Figure 3 molecules-28-01693-f003:**
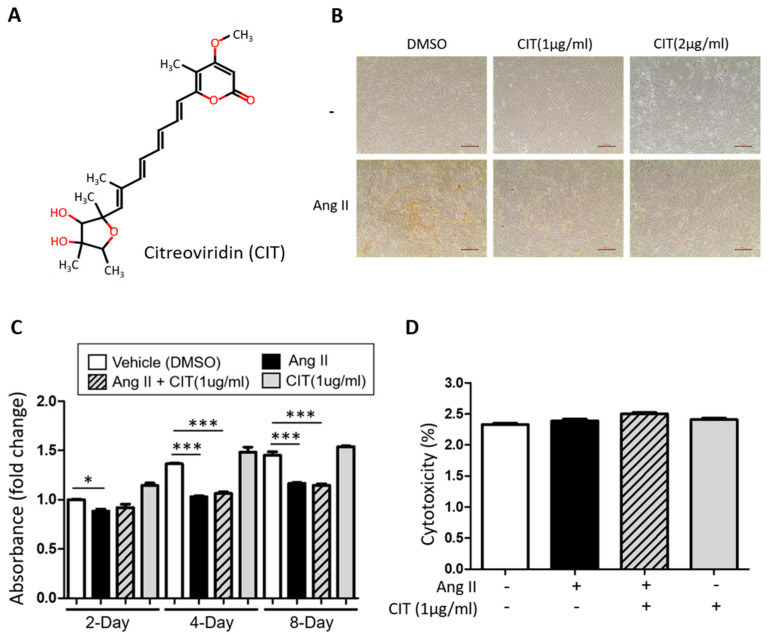
Citreoviridin suppresses vascular calcification without cell death. (**A**) The chemical structure of citreoviridin (CIT). (**B**) The morphology of HAoSMCs treated with citreoviridin was observed under a light microscopy. Scale bar: 500 μm. (**C**) The WST-1 assay and (**D**) cytotoxicity (LDH) on HAoSMCs were assessed in presence of Ang II with or without citreoviridin on 8 days. “+” means with Ang II or CIT and “-” means without Ang II or CIT. Quantitative data were presented as the mean ± SEM (*n* ≥ 3). * *p* < 0.05; *** *p* < 0.001.

**Figure 4 molecules-28-01693-f004:**
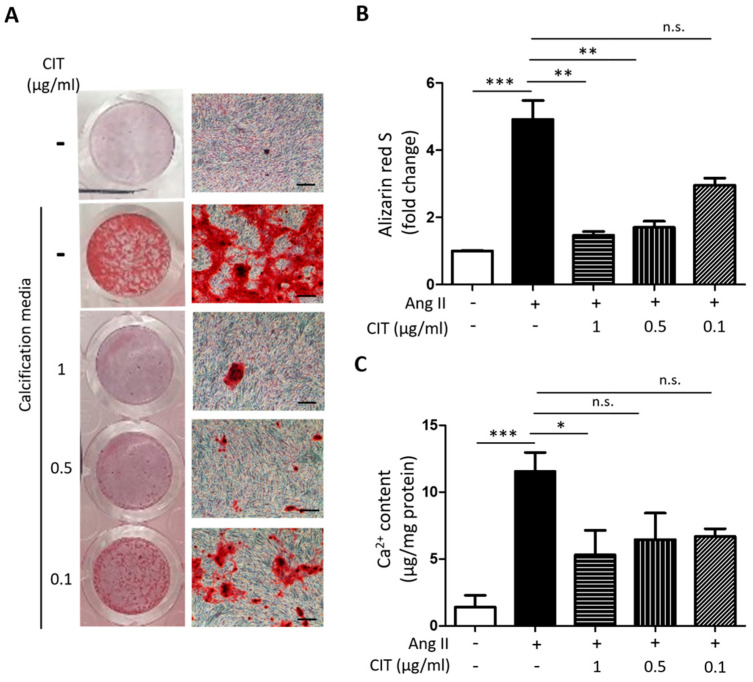
Citreoviridin attenuates the Ang II-induced vascular calcification. (**A**) Calcium deposition was assessed by alizarin red S staining and (**B**) the bar graph. Scale bar: 500 μm. (**C**) Calcium contents were evaluated under various concentrations of citreoviridin on Ang II-stimulated HAoSMCs. “+” means with Ang II and “-” means without Ang II or CIT. Data are shown as the mean ± SEM (*n* ≥ 3). * *p* < 0.05; ** *p* < 0.01; *** *p* < 0.001; n.s., not significant.

**Figure 5 molecules-28-01693-f005:**
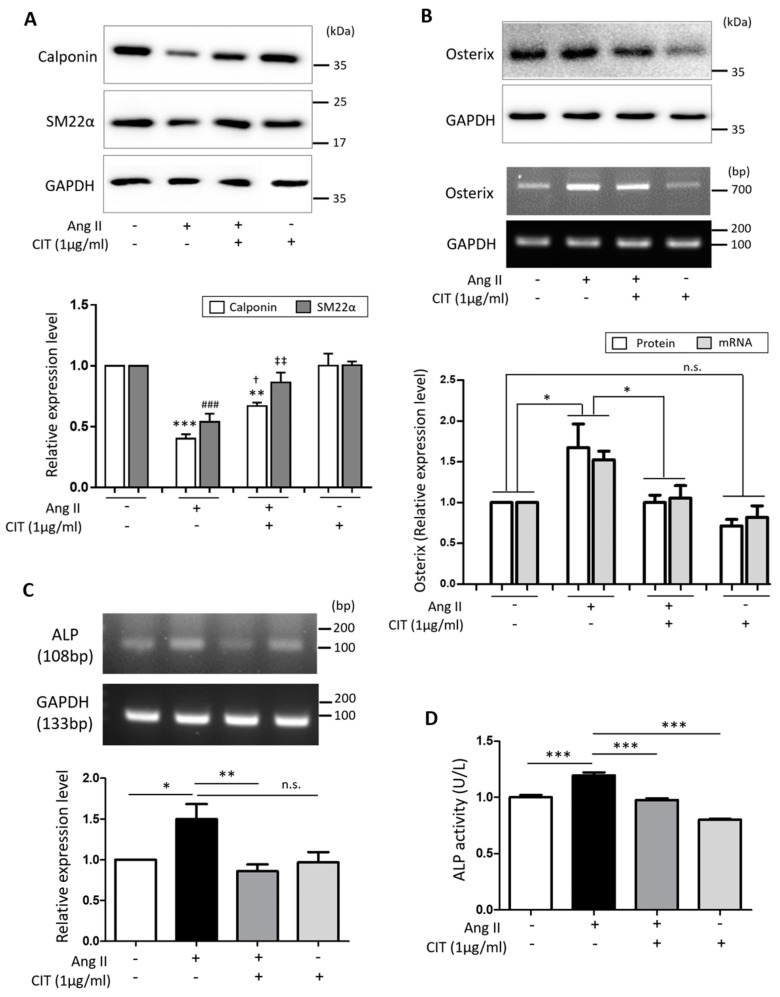
Citreoviridin attenuates Ang II-induced differentiation of HAoSMCs to osteogenic cells. (**A**) The expression level of smooth muscle cell markers, Calponin and smooth muscle protein 22-alpha (SM22α), were examined by Western blot. ** *p* < 0.01, *** *p* < 0.001 vs. control of Calponin; ^†^ *p* < 0.05 vs. Ang II of Calponin; ^###^ *p* < 0.001 vs. control of SM22α; ^‡‡^ *p* < 0.01 vs. Ang II of SM22α. (**B**) The expression level of Osterix as osteoclast specific marker by Ang II with or without citreoviridin were confirmed by Western blot and RT-PCR. In addition, the expression level and activity of ALP in (**C**) cell lysates and (**D**) supernatant of HAoSMCs were confirmed by RT-PCR and ALP assay. Western blot and RT-PCR results were normalized by GAPDH. “+” means with Ang II or CIT and “-” means without Ang II or CIT. Data are presented as the mean ± SEM (*n* ≥ 3). * *p* < 0.05; ** *p* < 0.01; *** *p* < 0.001; n.s., not significant.

**Figure 6 molecules-28-01693-f006:**
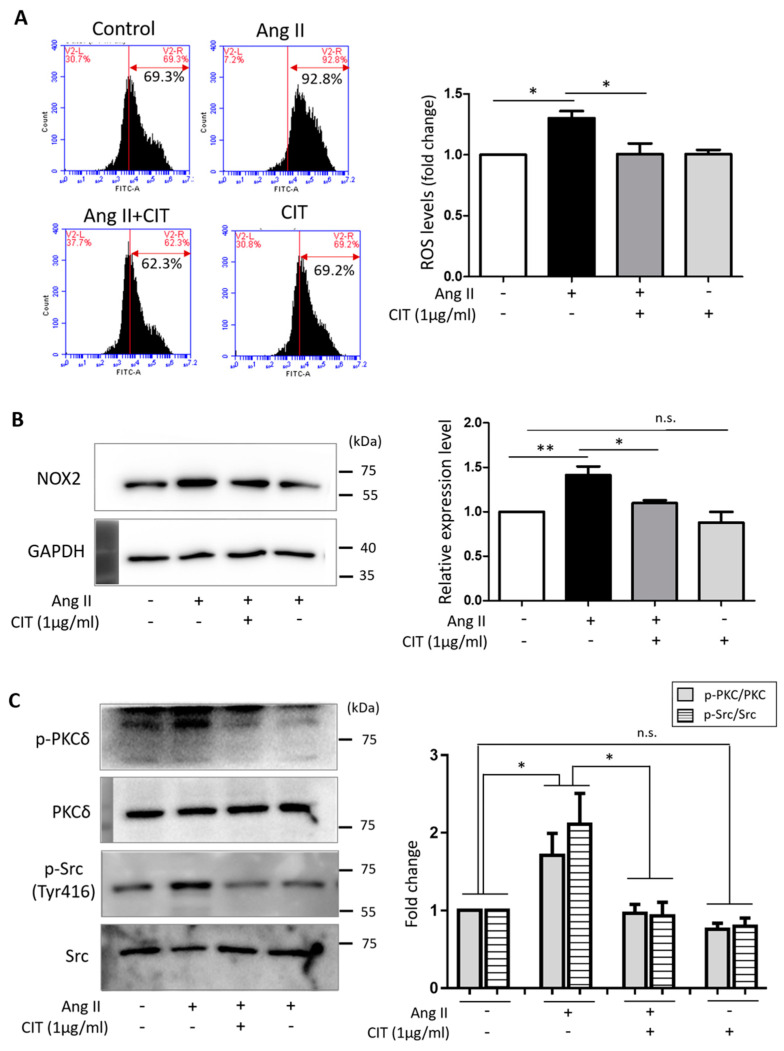
Citreoviridin decreased Ang II-induced ROS production in HAoSMCs. (**A**) Flow cytometry was conducted under Ang II with or without citreoviridin in HAoSMCs. (**B**,**C**) Western blotting showed that citreoviridin was decreased NOX2 protein level and phosphorylation levels of PKCδ and Src. NOX2 expression level was normalized by GAPDH. The relative values of phosphorylation levels of PKCδ and Src were normalized to the expression of PKCδ and Src. “+” means with Ang II or CIT and “-” means without Ang II or CIT. Data are presented as the mean ± SEM (*n* ≥ 3). * *p* < 0.05; ** *p* < 0.01; n.s., not significant.

**Figure 7 molecules-28-01693-f007:**
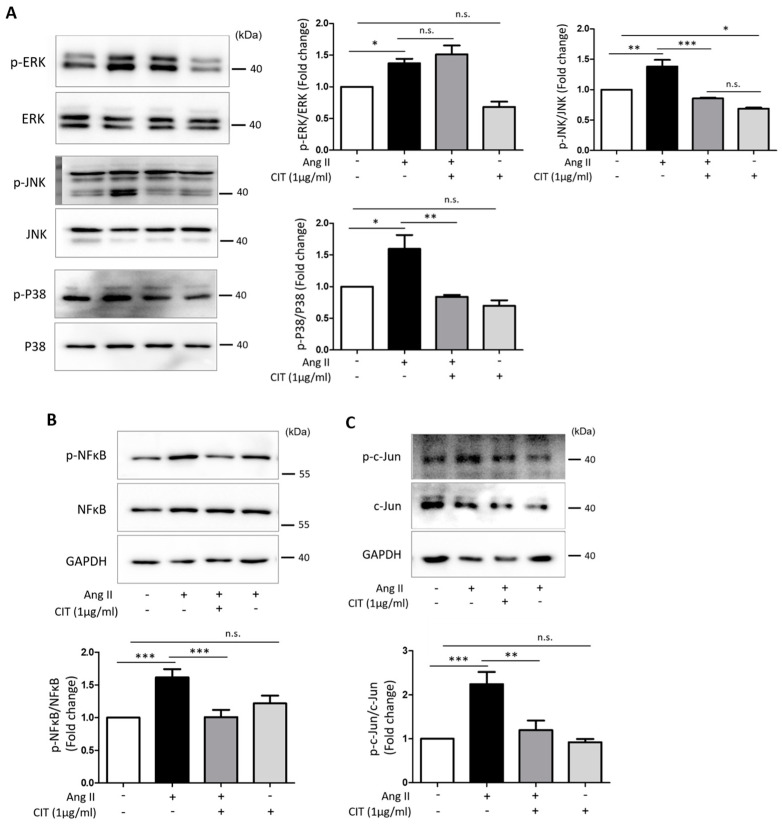
The inhibitory effect of phosphorylation of ERK, p38, JNK, NF-κB and c-Jun by citreoviridin in Ang II-stimulated HAoSMCs. (**A**–**C**) Phosphorylation of ERK, JNK, p38, NF-κB and c-Jun were confirmed by Western blot. The relative values of phosphorylation levels were normalized to the expression of ERK, JNK, p38, NF-κB and c-jun as an AP-1 family member. “+” means with Ang II or CIT and “-” means without Ang II or CIT. Data are shown as mean ± SEM (*n* ≥ 3). * *p* < 0.05; ** *p* < 0.01; *** *p* < 0.001.

**Figure 8 molecules-28-01693-f008:**
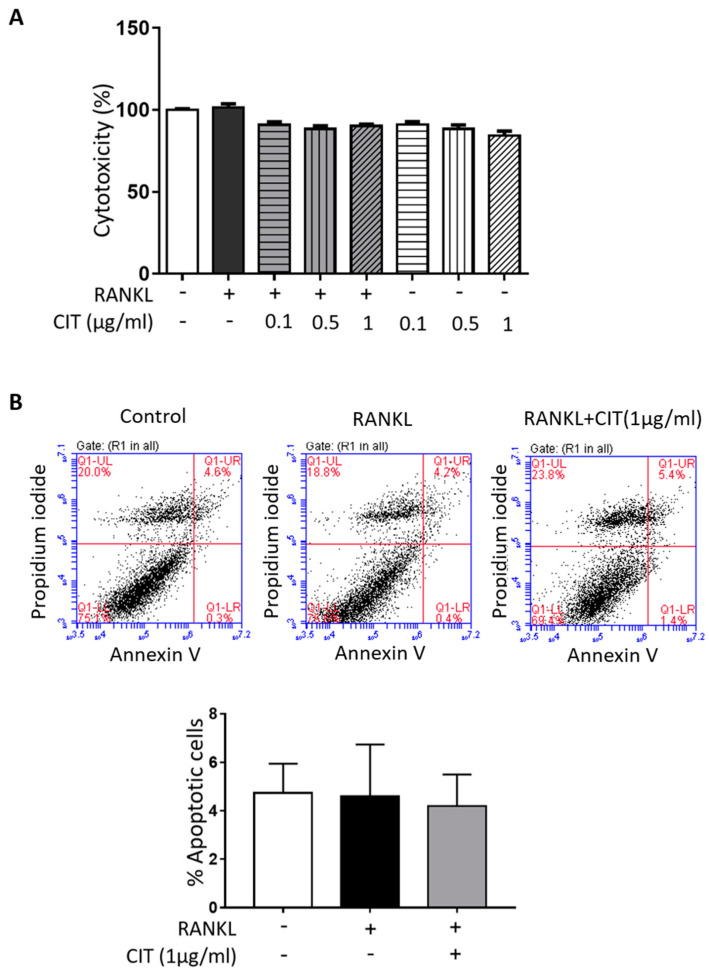
Citreoviridin inhibits osteoclast differentiation without cell death on Raw264.7 cells. (**A**) Amount of released LDH from Raw264.7 cells was measured after treatment with various concentration of citreoviridin (0.1, 0.5 and 1 μg/mL) in the presence or absence of RANKL. (**B**) The apoptosis rate of Raw264.7 cells was analyzed by flow cytometry with Annexin V/PI double staining after citreoviridin treatment. The upper right zone presents the late stage of apoptotic cells (Annexin V and PI positive). “+” means with RANKL or CIT and “-” means without RANKL or CIT. Data are presented as means ± SEM (*n* ≥ 3).

**Figure 9 molecules-28-01693-f009:**
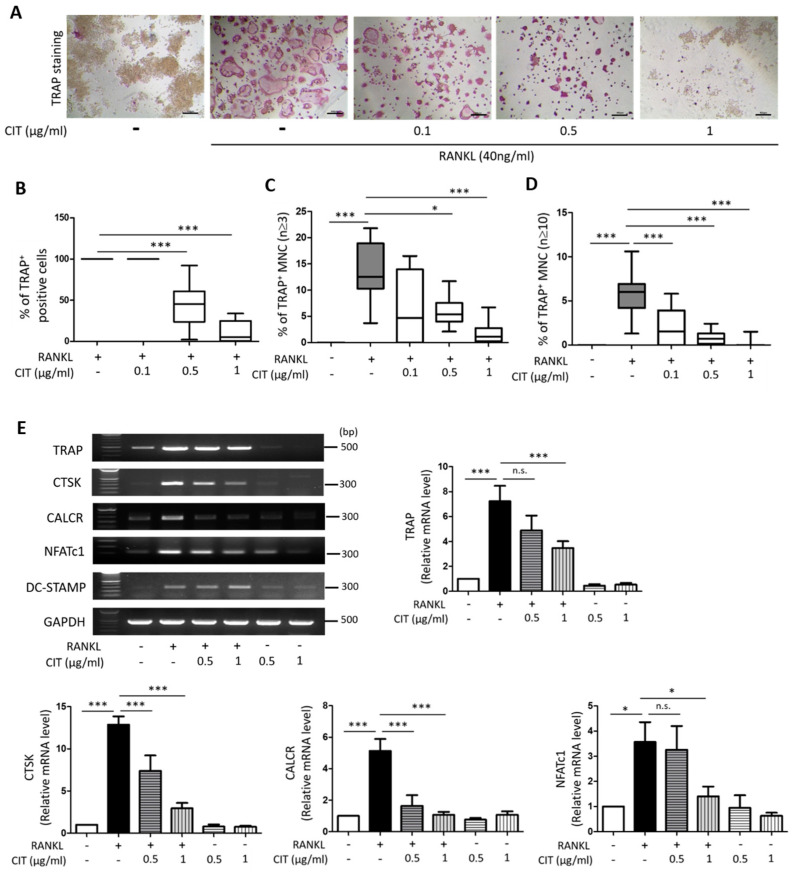
Citreoviridin inhibits the osteoclastic differentiation. (**A**) Raw264.7 cells were treated with various concentration of citreoviridin. Fixed cells were stained for TRAP and observed under a light microscope (40×). Scale bar = 400 μm. (**B**) Numbers of TRAP-stained cells, (**C**) TRAP-positive multinuclear cells (3 or more nuclei) and (**D**) 10 or more TRAP-positive multinuclear cells were counted in Raw264.7 cells treated with RANKL with or without Citreoviridin and then presented as percentages. (**E**) mRNA levels of osteoclast specific genes were examined by RT-PCR. The RT-PCR results were normalized to GAPDH about TRAP, CTSK, CALCR, NFATc1, and DC-STAMP (*n* ≥ 3). * *p* < 0.05; *** *p* < 0.001; n.s., not significant. “+” means with RANKL or CIT and “-” means without RANKL or CIT.

**Figure 10 molecules-28-01693-f010:**
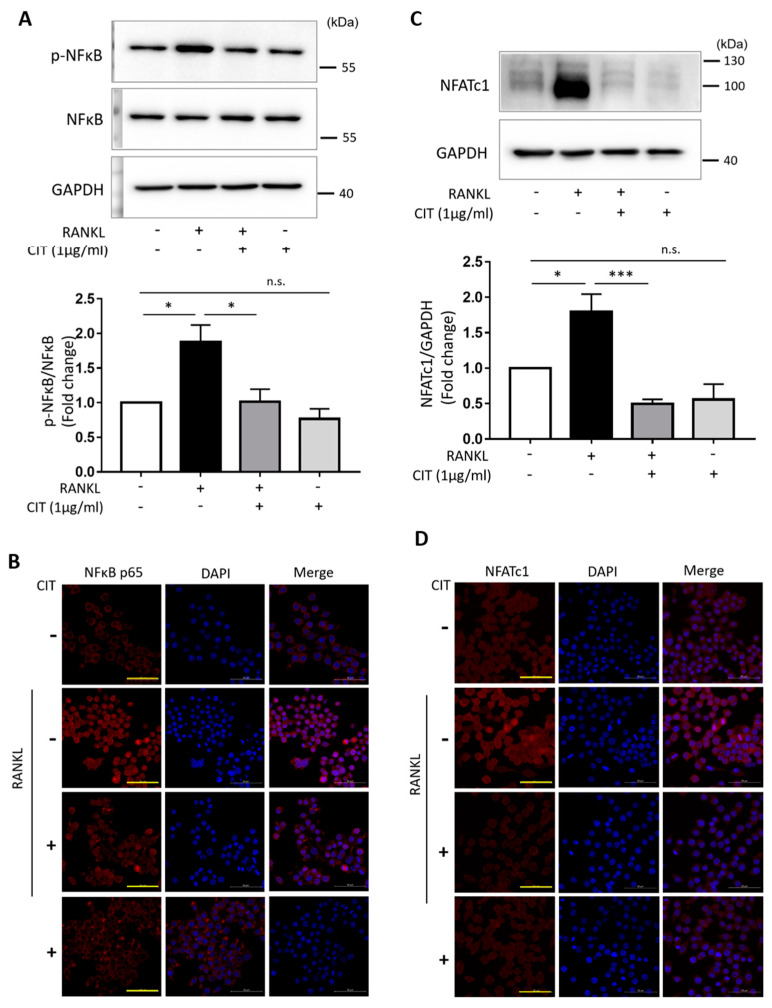
Citreoviridin regulates NF-κB-NFATc1 signals. (**A**) Protein expression and phosphorylation levels of NF-κB and (**C**) expression level of NFATc1 were examined in Raw264.7 cells treated with or without citreoviridin under RANKL stimulation. The phosphorylation level of NF-κB was normalized to respective total NF-κB, and NFATc1 expression level was normalized by GAPDH. (**B**,**D**) Effects of citreoviridin on NF-κB and NFATc1 translocation in Raw264.7 cells. Cells were examined to assess the translocation of NF-κB and NFATc1 into nucleus. The images (40×) were observed by using a LSM 700 laser scanning confocal microscope. Scale bar = 50 μm. Data are presented as the mean ± SEM, (*n* ≥ 3). * *p* < 0.05; *** *p* < 0.001; n.s., not significant. “+” means with RANKL or CIT and “-” means without RANKL or CIT.

**Figure 11 molecules-28-01693-f011:**
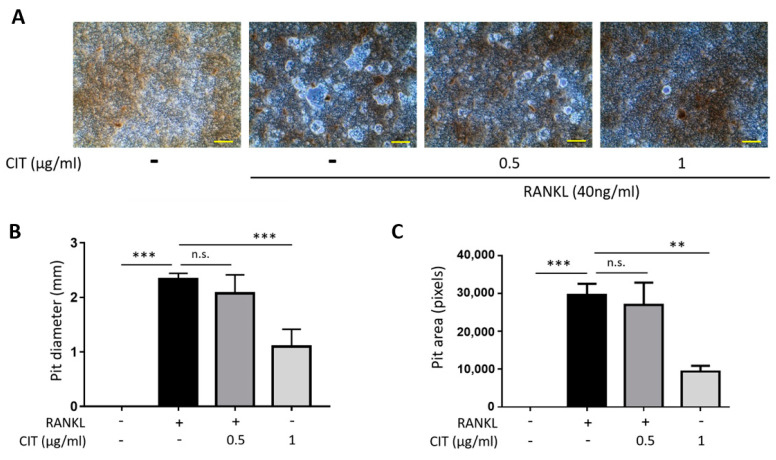
Citreoviridin attenuates RANKL-induced bone resorption. Raw264.7 cells were seeded on a CaP-coated plate with collagen type I and then stimulated by RANKL to induce differentiation to osteoclast. (**A**) Resorption pit areas were visualized by a light microscopy (100×). Scale bar = 400 μm. (**B**,**C**) Resorption pit areas were measured by mm scale and pixels. ** *p* < 0.01; *** *p* < 0.001; n.s., not significant. “+” means with RANKL and “-” means without RANKL or CIT.

**Figure 12 molecules-28-01693-f012:**
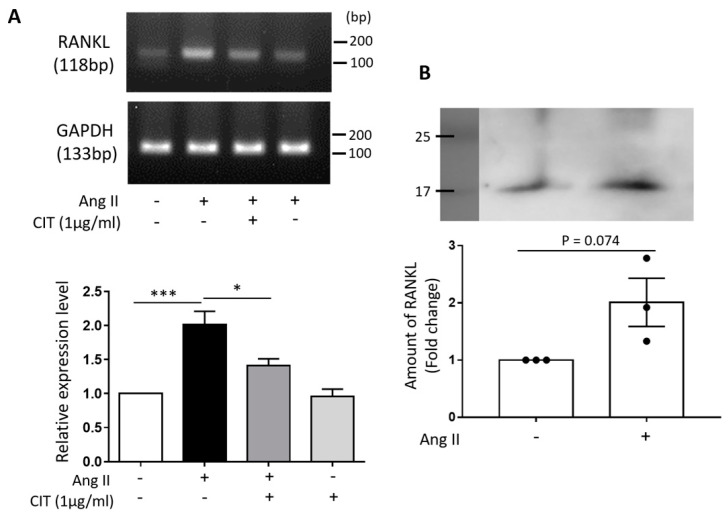
The expression of RANKL by Angiotensin II in HAoSMCs. Ang II was treated with HAoSMCs and the expression level of RANKL was confirmed by RT-PCR and Western blot. (**A**) The expression level of RANKL was assessed by RT-PCR using cell lysates treated by Ang II with or without citreoviridin. (**B**) The protein level of RANKL in cell supernatant was examined by Western blot. Results were normalized to GAPDH. “+” means with Ang II or CIT and “-” means without Ang II or CIT. Data are shown as the mean ± SEM (*n* ≥ 3). * *p* < 0.05; *** *p* < 0.001.

## Data Availability

Data are available in this article.

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
