# Peer review of "A Low Concentration of Citreoviridin Prevents Both Intracellular Calcium Deposition in Vascular Smooth Muscle Cell and Osteoclast Activation In Vitro"

_molecules, 2023, doi:10.3390/molecules28041693_

Round 1
Reviewer 1 Report
It is a good manuscript. I have two comments:
1. About the compound's toxicity, I think the toxicity reported in the literature should be included in the discussion of the result and not in conclusion. Include a comparative table about this.
2. looks weak the criteria of compound selection. Very similar compounds of natural origin exist; why do these not?
Author Response
19, January, 2023
Dear Editor,
Enclosed is our revised manuscript (molecules-2157084), “A low concentration of citreoviridin prevents both intracellular calcium deposition in vSMC and osteoclast activation in vitro” which we are submitting for publication in your respected journal, Molecules.
We authors thank you for your kind letter and are thankful for the reviewers’ constructive and critical comments concerning our manuscript. We have carefully studied the reviewer’s comments, and made necessary revisions, which we hope to solve the issues raised by the reviewers. We believe that our manuscript has been improved thanks to the reviewer’s comments, and the subject of this study meets the interest of your readership.
This work is not under consideration by any other publication nor has it been published in part or as a whole in any other journal. All authors have approved the manuscript and have agreed to its submission to your esteemed journal. All authors declare that there are no conflicts of interest.
Our responses to the comments given by the reviewer are as follows:
*Open Review1
Comments and Suggestions for Authors
- About the compound's toxicity, I think the toxicity reported in the literature should be included in the discussion of the result and not in conclusion. Include a comparative table about this.
Response to reviewer: With all due respect, frankly, we authors are afraid that we did not clearly understand the reviewer’s comments. First of all, the toxicity of CIT reported in the literature was described in the second paragraph of the Discussion, not in conclusion as the reviewer suggested. Second, we did not understand the meaning of making a comparative table. Does the reviewer want us to make a table for literatures reported toxicity of CIT? If so, since this is not a review paper, we authors believe that making a table for already published literatures is kind of unusual. If we authors completely misunderstood the reviewer’s intention, please give us more detailed direction on the comparative table so that we can more properly address the issue.
- looks weak the criteria of compound selection. Very similar compounds of natural origin exist; why do these not?
Response to reviewer: Assuming the first sentence and the second one are asking one question, we authors understood the reviewer’s comments as “there are other natural compounds very similar to CIT, but you selected CIT, why is that?” If that was indeed the reviewer’s question, the answer is that the main purpose of this study was to find a compound effective for both VC and osteoporosis using a certain natural product library, not to examine the biological role of CIT and similar compounds such as mycotoxin. In other words, if the main purpose of this study was to examine the biological role of bacteria-derived compounds, we would have examined and compared CIT and similar natural compounds as the reviewer suggested. However, the starting point of this study was the VC and osteoporosis, not bacteria-derived natural compounds including CIT. CIT just happens to be in the natural product library we used, and it was effective for both conditions. Therefore, we did not examine other similar natural products for this study. However, the reviewer’s question raised an interesting point, and comparing the effects of other similar compounds on VC and osteoporosis will make an interesting subject for our future study. We authors deeply appreciate the time and efforts the reviewers put into to further improve our manuscript.
Reviewer 2 Report
Minor comments in the pdf attached

Author Response
19, January, 2023
Dear Editor,
Enclosed is our revised manuscript (molecules-2157084), “A low concentration of citreoviridin prevents both intracellular calcium deposition in vSMC and osteoclast activation in vitro” which we are submitting for publication in your respected journal, Molecules.
We authors thank you for your kind letter and are thankful for the reviewers’ constructive and critical comments concerning our manuscript. We have carefully studied the reviewer’s comments, and made necessary revisions, which we hope to solve the issues raised by the reviewers. We believe that our manuscript has been improved thanks to the reviewer’s comments, and the subject of this study meets the interest of your readership.
This work is not under consideration by any other publication nor has it been published in part or as a whole in any other journal. All authors have approved the manuscript and have agreed to its submission to your esteemed journal. All authors declare that there are no conflicts of interest.
Our responses to the comments given by the reviewer are as follows:
Open Review2
Thank you for the thorough and valuable comments. We highlighted the corrected part yellow in the revised manuscript.
As we briefly summarize the improvements,
- All 3 supplementary figures are included in the main manuscript.
- Detailed information of every materials such as a natural compound library which we used is described in Materials and Methods.
- We authors added the full name on every first mentioned abbreviations.
- Every first mentioned manufacturer have detailed information (city and country).
- In author contribution, among the all authors, “Soyeon Lim and Seahyoun Lee” have modified initial such as “S. Lim and S. Lee” unlike other authors, otherwise it is hard distinguish between 2 authors (S.L vs. S.L.).
- Number of experiments was described at the end of every Figure legends.
- Lines of references are modified.